# What is coastal subsidence?

Torbjörn E. Törnqvist[1] and Michael D. Blum[2]

[1]Department of Earth and Environmental Sciences, Tulane University, 6823 St. Charles Avenue, New Orleans, Louisiana 70118-5698, USA and [2]Earth, Energy and Environment Center, The University of Kansas, 1414 Naismith Drive, Lawrence, Kansas 66045, USA

## Perspective

vertical land motion; subsidence; coast; sea-level rise

**Corresponding author:**
Torbjörn E. Törnqvist;
Email: tor@tulane.edu

### Abstract

Major technological advances have made measurements of coastal subsidence more sophisticated, but these advances have not always been matched by a thorough examination of what is actually being measured. Here we draw attention to the widespread confusion about key concepts in the coastal subsidence literature, much of which revolves around the interplay between sediment accretion, vertical land motion and surface-elevation change. We attempt to reconcile this by drawing on well-established concepts from the tectonics community. A consensus on these issues by means of a common language can help bridge the gap between disparate disciplines (ranging from geophysics to ecology) that are critical in the quest for meaningful projections of future relative sea-level rise.

### Impact statement

Coastal subsidence is increasingly recognized as a major threat for the rapidly growing populations in coastal lowlands worldwide. Monitoring and predicting coastal subsidence is challenging and requires a multitude of methods. As a result, researchers within this community have diverse backgrounds, which has resulted in differences in the conceptual framework for understanding this process, including differences in terminology and definitions. This paper proposes a unifying framework to understand coastal subsidence with the goal to enhance communication between these different research communities.

Land subsidence, defined herein as the downward motion of a specific reference horizon relative to a fixed datum, is a major compounding problem for low-elevation coastal zones that are feeling the effects of accelerating global sea-level rise, including some that host the world's largest population centres. This "slow-motion disaster" is receiving rapidly increasing attention within the research community (Buffardi and Ruberti, 2023) and is prevalent along depositional coastlines that are subject to long-term passive margin subsidence. Superimposed on this relatively steady ($10^5$ to $10^6$ yr or more) and slow geologic process are non-steady components of vertical land motion (VLM), including glacial isostatic adjustment that affects every shoreline worldwide over shorter timescales ($10^3$ to $10^5$ yr), plus compaction of recently deposited sediment as well as compaction due to fluid extraction that operate over timescales as short as $10^0$ to $10^2$ yr and exhibit large spatial variability but often dominate subsidence on local scales. Here we argue that the complexities of coastal subsidence are not always recognized and appreciated within the multidisciplinary subsidence research community, presenting an obstacle to subsidence projections and coastal policymaking. We therefore ask the simple question "What is coastal subsidence?"

Shirzaei et al. (2021) argued that within the context of VLM, distinction must be made between static and dynamic coastal landscapes. We use these terms in a morphodynamic sense: in the former, deposition and erosion have been essentially halted, whereas in the latter (e.g., coastal wetlands), these processes operate relatively uninhibited. Morphodynamically "static" landscapes include urban as well as agricultural settings with minimal relief, which are common in low-elevation coastal zones and characterized by geomorphic processes that are sufficiently slow that they can be neglected over human-relevant timescales. The implications of this distinction for subsidence measurements are profound: in static landscapes, VLM can be directly observed by measurements of land surface displacement with remote sensing and geodetic techniques. Therefore, subsidence studies in static settings are often relatively straightforward. The situation is entirely different in dynamic landscapes, where such methods measure surface-elevation change (SEC) rather than VLM (Figure 1). These are fundamentally different things.

To progress towards a common understanding, we consider VLM and SEC in subsiding, coastal depositional landscapes to be the mirror image of what occurs in uplifting, erosional

landscapes, as outlined in the seminal paper of England and Molnar (1990) where rock uplift was differentiated from surface uplift. Rock uplift is the upward VLM of a specific reference horizon relative to a fixed datum, whereas surface uplift equals rock uplift minus erosion (exhumation). Surface uplift is therefore one variant of SEC as used herein, but if there is no erosion, or no deposition for that matter, rock uplift equals SEC. Subsidence is the opposite of rock uplift and represents the downward VLM (a negative number) of a specific reference horizon relative to a fixed datum, and SEC equals the difference between VLM of the reference horizon and vertical accretion (or erosion) of sediment. SEC will be a positive number if vertical accretion exceeds subsidence, SEC will equal zero if vertical accretion balances subsidence, and SEC will be negative if subsidence exceeds vertical accretion. As such, subsidence is defined in a way that is consistent with its long-standing use in the stratigraphic literature, where subsidence measurements can be referred to any horizon in the subsurface regardless of depth or timescale (e.g., Paola, 2000; Frederick et al., 2019).

Measuring SEC and/or VLM can be accomplished by a wide range of methods, some of which strictly determine changes at the land surface, whereas others explicitly monitor subsurface processes. Combining both is essential to disentangling driving mechanisms. Space-based methods like Interferometric Synthetic Aperture Radar (InSAR) have become some of the most powerful tools to obtain spatially continuous data on SEC and/or VLM (e.g., Jones et al., 2016). Global Navigation Satellite System (GNSS) data can provide point observations on VLM in coastal settings (e.g., Hammond et al., 2021) and are often used to ground truth InSAR measurements. While InSAR is potentially invaluable, it cannot differentiate the drivers, or causal mechanisms, if multiple processes contribute to SEC. Most importantly, InSAR fundamentally measures SEC, which will be equivalent to VLM in static landscapes only (Figure 1). Even in such settings, care must be taken that reflectors measure change at the land surface, given that large buildings often rest on foundations that may be tens of metres deep. As a result, significant differences may exist between velocities obtained from reflectors associated with buildings that rest on deep pilings compared to adjacent urban infrastructure, where the latter often exhibits higher rates (De Wit et al., 2021).

Despite this caveat, ground truthing of InSAR data with GNSS measurements is comparably straightforward in static landscapes (e.g., Fabris et al., 2022). In contrast, as discussed by Keogh and Törnqvist (2019), this may be more challenging in wetland environments where subsidence in the shallowest portion of the subsurface is typically not captured by GNSS instruments (Figure 1). As a result, understanding subsidence in such settings requires

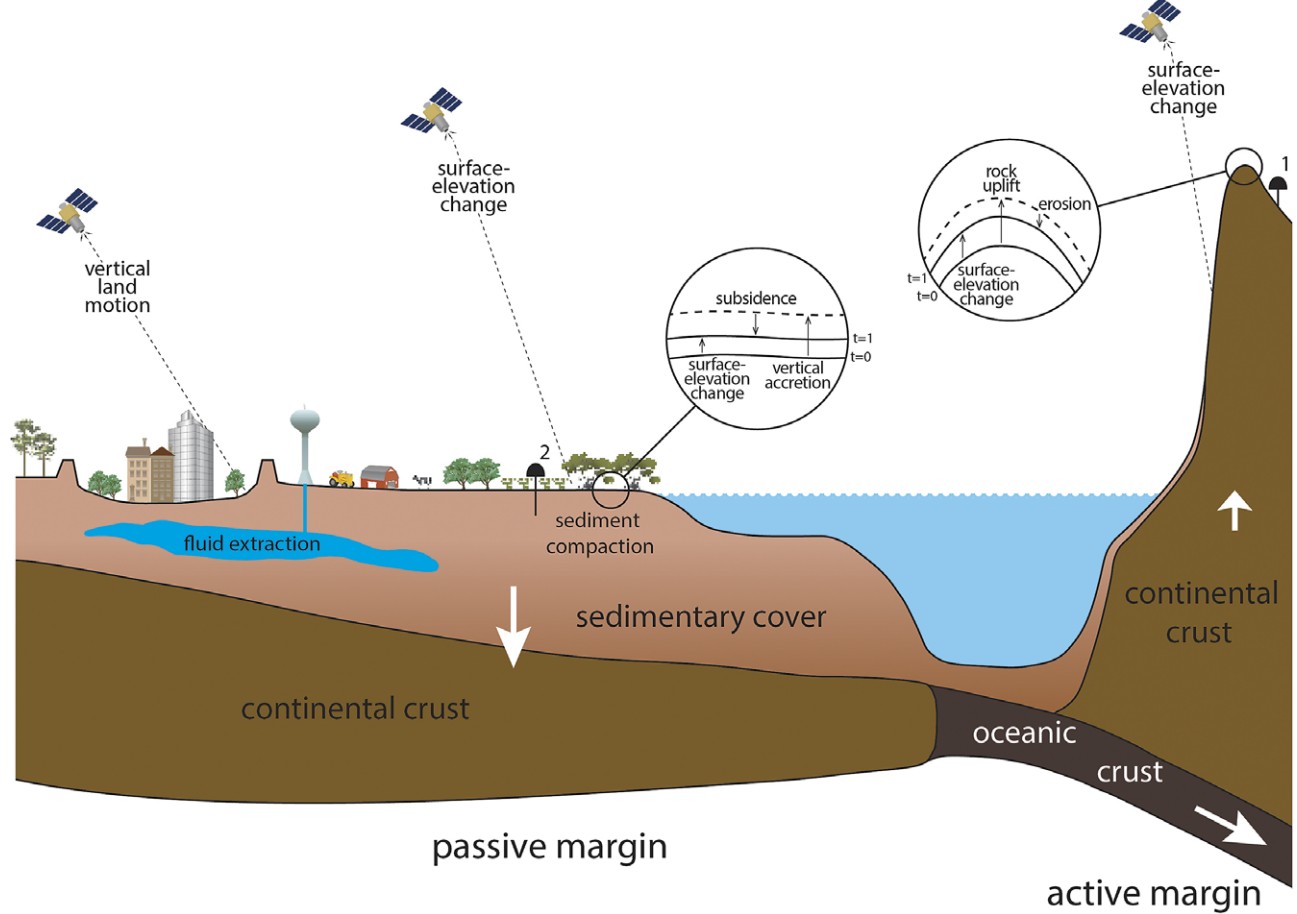

**Figure 1.** Schematic cross section of two continental margins subject to subsidence (passive margin) and uplift (active margin), respectively. Note that InSAR measurements provide rates of surface-elevation change, which can be directly interpreted as vertical land motion only in static landscapes (e.g., urban areas). Also note the difference between the two GNSS stations: #1 is set in exposed bedrock and measures rock uplift, whereas #2 measures subsidence but misses the shallow compaction component because the instrument rests on a foundation several meters below the wetland surface. Insets (circles) provide details of the key processes operating in erosive uplifting settings, versus subsiding coastal wetlands where deposition can drive shallow sediment compaction. In the latter, there is subsidence despite net surface-elevation gain.

independent measurements of vertical accretion and/or erosion to complement time series on SEC from InSAR. (Separate from these considerations, it should be noted that the collection of InSAR data in wetlands is challenging – a topic beyond the scope of the present paper.) It is therefore imperative that vertical accretion not be used synonymously with SEC as has been the case in recent, widely cited papers (e.g., Crosby et al., 2016; FitzGerald and Hughes, 2019). Along the same lines, InSAR measurements that encompass both static and dynamic landscapes (e.g., Ohenhen et al., 2023) measure SEC, which can only be equated with VLM in the static case.

A related, widespread source of confusion is associated with the interplay between deposition and subsidence. For example, studies of river delta vulnerability have implied that a reduction of sediment supply increases subsidence (e.g., Becker et al., 2020; Glover et al., 2023). However, the opposite is true: vertical accretion and subsidence from compaction in the upper portion of the sediment column are closely coupled (Saintilan et al., 2022) due to the effective stress exerted by newly accumulated sediment (Zoccarato and Da Lio, 2021). In other words, the addition, not the reduction of, sediment will therefore increase subsidence. Nevertheless, even though deposition typically enhances subsidence, it still often results in net surface-elevation gain (Chamberlain et al., 2021; Saintilan et al., 2022). Put differently, wetlands that lose elevation compared to rising sea level may still gain elevation with respect to a fixed geodetic datum, as long as vertical accretion outpaces subsidence (Figure 1).

Why is this important? We argue that not recognizing these nuances could hinder progress, not just scientifically but also in terms of the policy implications of coastal subsidence. For example, it is not uncommon for river deltas to exhibit positive SEC due to ongoing sediment deposition. Data presented by Jankowski et al. (2017) show that the rate of SEC in coastal Louisiana is $0.7 \pm 6.9$ mm/yr. This is due to vertical accretion that largely offsets subsidence (58% of their monitoring sites exhibit values of zero or higher). However, none of this means that subsidence is not occurring throughout the shallow and deep subsurface; Nienhuis et al. (2017) reported a subsidence rate averaging 9 mm/yr for this area. Put differently, conflating subsidence with SEC in a case like this would make it very challenging to effectively communicate the subsidence problem in this region to practitioners. Given the increasing need for predictive, process-based subsidence models (Allison et al., 2016; Shirzaei et al., 2021), it is critical to separate between elevation loss due to downward VLM (i.e., subsidence) and elevation loss due to a lack of sediment deposition or erosion, in the same way that tectonic geomorphologists separate between rock uplift and surface uplift (e.g., Gasparini and Whipple, 2014; Yang et al., 2015). We advocate a similar theoretical framework for the coastal subsidence community.

An important motivation for this contribution is the increasing recognition of coastal subsidence as an existential threat that adds to the risks posed by global sea-level rise for millions of people worldwide. In fact, along many deltaic coastlines the magnitude of coastal subsidence can equal or exceed current and projected rates of geocentric sea-level rise (Jelgersma, 1996; and many subsequent studies). In this sense, coastal subsidence research has enjoyed rapid progress, not least due to a wide range of technological advances and an increasingly high spatial and temporal resolution in the detection of VLM at or near the Earth's surface (e.g., Da Lio et al., 2018; Steckler et al., 2022; Zoccarato et al., 2022; Zumberge et al., 2022). However, as shown by the examples discussed above, new and/or increasingly sophisticated measurements have not always gone hand in hand with progress on our understanding of the relevant surface and subsurface processes: the vital question of "what exactly is being measured?" must never become an afterthought.

In closing, we note that increased concern about subsidence will be addressed by the recently established International Panel on Land Subsidence (IPLS; Minderhoud and Shirzaei, 2022; https://sites.google.com/view/iplsubisdence/home). One of the key objectives of the IPLS will be to produce subsidence projections that can be combined with IPCC-style sea-level projections (Oppenheimer et al., 2019; Fox-Kemper et al., 2021) to generate more powerful forecasts of relative sea-level change. A recent community paper on sea-level terminology (Gregory et al., 2019) has reduced confusion on critically important concepts from this neighbouring field. In that spirit, we hope that this brief paper can be an initial contribution towards a more clearly defined conceptual framework for understanding and projecting subsidence in the coastal zone.

**Open peer review.** To view the open peer review materials for this article, please visit http://doi.org/10.1017/cft.2024.1.

**Acknowledgements.** Jaap Nienhuis and an anonymous reviewer provided numerous thoughtful comments that enabled us to clarify key elements of the subject matter, but we stress that the opinions expressed herein are our own. We also thank Philip Minderhoud and Nicole Gasparini for valuable feedback.

**Competing interest.** The authors declare none.

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
