## [Reviewer Report]

Tornqvist and Blum present on a widespread terminology issue surrounding VLM, subsidence, and surface elevation change. I find it well-written and easy to read. I fully agree that a better conceptual framework would benefit the diverse subsidence community, most importantly any new student entering this field.

I have one comment, and that is, in my view, that there is not always a “miscommunication” (L12) about subsidence, but that part of the issue is simply that it is approached differently in different scientific disciplines. As far as I understand, the distinction is that for the geodesy community, sedimentation/erosion is part of SLR, whereas in geomorphology it is not. Although I am very much part of the geomorphology community and I follow the same definitions as Tornqvist and Blum, I would not say that either science discipline is wrong.

This comes back in several instances in their paper. For example, in L71, perhaps Becker et al and Glover et al follow the geodesy definitions. Then this is not necessary a misconception and a reduction in sediment supply (resulting in erosion) would lead to “subsidence”? In another instance, when reading the Gregory et al paper that Tornqvist and Blum cite (L94), it seems to me that these authors (Gregory) also follow the geodesy definition when they write: “The level of the sea floor varies due to solid-Earth tides, accumulation of sediment (with eventual compaction) and vertical land movement on a range of timescales.” They differentiate consolidated and unconsolidated sediments but they do not specify a degree of consolidation.

Note that in a recent paper of mine (Nienhuis et al., Annual Reviews 2023, co-authored by Tornqvist), we also briefly discussed this point. Here we wrote that “some studies consider sedimentation and erosion to be contributors to VLM and, therefore, RSL (e.g., Dalca et al. 2013). Here we follow the geomorphologic literature and consider sedimentation and erosion to be separate from VLM such that surface-elevation change is the sum of VLM, sedimentation, and erosion.”

To make a long story short: I fully agree with the content of the paper and I think it has great merit in the scientific literature. One aspect I would personally like to see added (or changed) is the fact that it may not be a “misconception” but rather a different use of terminology stemming from different scientific disciplines. This, of course, does not imply that the terminology is used correctly in every paper and there may yet be many studies where subsidence/VLM/SEC was actually used incorrectly.

Two minor points:

L59; perhaps add why the foundation depth is relevant in this case (i am aware this is because of vertical changes in the subsidence rate, but this may not be obvious to all readers).

L68: be equated with VLM in [the static case]

Best wishes,

Jaap Nienhuis

---

## [Reviewer Report]

Dear Tor and Michael,

I read your ms with interest. I agree that a clarification on the subject and the related terminology is really needed. I would like to add my point of view, i.e, that of an engineer and/or hydrogeologist, to this contribution with the hope of enlarging the “validity” of the definitions. In fact, in the present form people/researchers with background like mine (which started working on the topic since the first decades of the last century) do not feel comfortable.

Two are the main issues:

1) the definition of “land subsidence” cannot differ from that of “land subsidence in coastal areas” (or coastal subsidence). In some why, the problem of “what is actually been measured” cannot rule the definition land subsidence. In other words, we cannot define what land subsidence is based on the monitoring techniques presently available. The techniques must adapt to the concept;

2) in the present form of the ms, the distinction between “displacements” and the “processes” responsible for these displacements is somehow lacking. There is a sort of mixture between the terminology used to represent “movements” and that used to represent “processes”. From the perspective of a reader aimed at disentangling / modelling (not only monitoring) the various physical processes causing the movements, this issue is worth to be clarified.

I think the ms should start with the definition of what “land subsidence” is. In the first paragraph (lines 30-44, p1) the term “land subsidence” is mentioned several times but without explaining what the word means (or what is the definition for the word). For example: “it is a compounding problem for low-elevated coastal zones (only?)”, or “long-term passive margin subsidence”, or “dominate subsidence on local scale”. But what is land subsidence? After the first paragraph the reader is really confused.

From an engineering point of view, “land subsidence” is the loss of elevation (with respect to a geocentric reference system, coherently with the sea-level rise) of the actual land surface, independently of all the processes that can superpose to generate the movement (and if the actual land surface gains elevation, we observe a “land uplift”) and irrespectively where this movement occurs (in “static” or “dynamic” landscapes). It is not the movement of a specific grain in the soil stratigraphy or the thickness loss/gain of a certain (shallow or deep) layer.

The variety of processes that superpose to cause land subsidence/uplift is large, ranging from deep and slow (tectonics), to intermediate (GIA, SIA), to (relatively) shallow and fast such as compaction due to subsurface fluid removal, shallow auto-compaction, erosion, oxidation, sedimentation, surface loading. And they can act independently on the site, in inner basins and coastal zones.

In view of understanding the fate of coastal area to future sea level rise, definitions such as “long-term passive margin subsidence” (p.1), “downward VLM of a specific reference horizon relative to a fixed datum” (p2), and “shallow subsidence” (figure 1) seem inconsistent and make confusion. Three different terms, the first and the latter referring to processes, the second one to the movement of a specific level.

Conversely, the “engineering” definition is consistent with what is reported in the last paragraph of your ms: “increasing recognition of coastal subsidence as an existential threat that adds in the risks posed by global sea level rise”. It is the movement of the actual land surface of a certain coastal area that must be considered when we aim at evaluating coastland resilience to sea level rise. It is not a matter of which displacement affect the geologic horizon dated 1000 yr BP (i.e., the VLM) or the thinness loss of the sediments deposited over the last 1000 yr (i.e., vertical accretion minus shallow subsidence, see Figure 1).

Indeed, a wetland that loose elevation with respect to a rising mean seal level may still uplifting or subsiding depending if processes contributing to gain elevation (e.g., deposition of new sediments, deep tectonics, GIA, or expansion of aquifers after well shutdown) prevails or not to processes causing elevation loss (such as sediment auto-compaction, compaction due to hydrocarbon exploitation, biodegradation of organic matter, deep tectonics, etc).

I agree that the outcomes of different monitoring systems must be properly interpreted depending on what they really measure. And this can differ in static and dynamic coastal landscapes. And a huge warning must be advanced, especially today when thousands of young scientists apply remote sensing without any or with a small knowledge of what they are measuring.

Please consider these notes in your ms. This way, the definition of land subsidence (uplift) is unique and has a general validity, independently on the study site (coastlands, inner plains, mountains). Eventually, at least make a clear distinction between the terminology related to processes and that related to (vertical) movement of surfaces. E.g., the term “shallow subsidence” seems wrong to me, while “shallow compaction” is unequivocal irrespective of the scientific background.

A few minor suggestions:

- l24 : “sediment compaction and fluid extraction” cannot be listed together. “Sediment autocompaction and compaction due to fluid extraction” are more consistent processes to be consecutively listed;

- l36: remote sensing techniques do not measure a “deformation” but “displacements”. Extensometers and fiber optics measure a deformation;

- l55 to l58: these lines seem a bit unclear and confuse the reader. Does GNSS differentiate drivers contributing to SEC? I don’t think so, GNSS has the same capability of InSAR, i.e. they both measure displacements. Only with GNSS antennas or radar reflectors founded at different depths give the possibility to differentiate the drivers. Moreover, it is reported that “InSAR measures SEC, which is equal to VLM in static landscapes”; but what about “land subsidence”? InSAR does not measure land subsidence? With these definitions the goal of defining what is land subsidence in coastal areas seems resolved by removing the word “land subsidence”. Similarly, in l67 to l69;

- l76 to l78: it is unclear to me what do you mean with “Wetlands that lose elevation with respect to a relative sea-level rise”. It seems a condition that refers to a “double references”: the sea that rises with respect to a certain land surface and the wetland that loses elevation with respect to this relative sea level movement. Very complex. Is it better to write “wetlands that lose elevation with respect to a rising sea-level”?

- figure 1: “shallow subsidence” is a wrong concept from my point of view, and it should be substituted with “shallow compaction”. Otherwise, there is the risk of increasing the misunderstanding about land subsidence instead of clarifying the topic.

---

## [Editor Report]

Dear Tor and Michael, 

Two reviewers have now seen your manuscript. As you see, both reviewers are positive about the manuscript and applaud its goals, but they also provide some useful pointers on how to further refine&clarify the definitions, in particular keeping in mind how there may be differences between various fields/backgrounds. I therefore return the manuscript to you for minor revisions. 

Kind regards, 

Aimée